# Methylation of *Salmonella* Typhimurium flagella promotes bacterial adhesion and host cell invasion

Julia A. Horstmann[1,2,13], Michele Lunelli ![ORCID][3,13], Hélène Cazzola[4], Johannes Heidemann[5], Caroline Kühne[6], Pascal Steffen[7], Sandra Szefs[1], Claire Rossi[4], Ravi K. Lokareddy[8], Chu Wang[3], Laurine Lemaire[4], Kelly T. Hughes[9], Charlotte Uetrecht ![ORCID][5,10], Hartmut Schlüter[7], Guntram A. Grassl[11], Theresia E. B. Stradal ![ORCID][2], Yannick Rossez[4], Michael Kolbe ![ORCID][3,12,14✉] & Marc Erhardt ![ORCID][1,6,14✉]

The long external filament of bacterial flagella is composed of several thousand copies of a single protein, flagellin. Here, we explore the role played by lysine methylation of flagellin in *Salmonella*, which requires the methylase FliB. We show that both flagellins of *Salmonella enterica* serovar Typhimurium, FliC and FljB, are methylated at surface-exposed lysine residues by FliB. A *Salmonella* Typhimurium mutant deficient in flagellin methylation is outcompeted for gut colonization in a gastroenteritis mouse model, and methylation of flagellin promotes bacterial invasion of epithelial cells in vitro. Lysine methylation increases the surface hydrophobicity of flagellin, and enhances flagella-dependent adhesion of *Salmonella* to phosphatidylcholine vesicles and epithelial cells. Therefore, posttranslational methylation of flagellin facilitates adhesion of *Salmonella* Typhimurium to hydrophobic host cell surfaces, and contributes to efficient gut colonization and host infection.

[1] Junior Research Group Infection Biology of Salmonella, Helmholtz Centre for Infection Research, Inhoffenstraße 7, 38124 Braunschweig, Germany. [2] Department of Cell Biology, Helmholtz Centre for Infection Research, Inhoffenstraße 7, 38124 Braunschweig, Germany. [3] Department for Structural Infection Biology, Center for Structural Systems Biology (CSSB) & Helmholtz Centre for Infection Research, Notkestraße 85, 22607 Hamburg, Germany. [4] Université de Technologie de Compiègne, Alliance Sorbonne Université, UMR7025 CNRS Enzyme and Cell Engineering Laboratory, Rue Roger Couttolenc, CS 60319, 60203 Compiègne, Cedex, France. [5] Heinrich Pette Institute, Leibniz Institute for Experimental Virology, Martinistraße 52, 20251 Hamburg, Germany. [6] Humboldt-Universität zu Berlin, Institute for Biology – Bacterial Physiology, Philippstr. 13, 10115 Berlin, Germany. [7] Institute of Clinical Chemistry and Laboratory Medicine, Mass Spectrometric Proteomics Group, University Medical Center Hamburg-Eppendorf, Martinistrasse 52, 20246 Hamburg, Germany. [8] Department of Biochemistry and Molecular Biology, Thomas Jefferson University, 1020 Locust Street, Philadelphia, Pennsylvania 19107, USA. [9] University of Utah, Department of Biology, Salt Lake City, UT 84112, USA. [10] European XFEL GmbH, Holzkoppel 4, 22869 Schenefeld, Germany. [11] Institute of Medical Microbiology and Hospital Epidemiology, Medizinische Hochschule Hannover, and German Center for Infection Research (DZIF), Partner Site Hannover-Braunschweig, Carl-Neuberg-Str. 1, 30625 Hannover, Germany. [12] MIN-Faculty University Hamburg, Rothenbaumchaussee 19, 20148 Hamburg, Germany. [13]These authors contributed equally: Julia A. Horstmann, Michele Lunelli. [14]These authors jointly supervised this work: Michael Kolbe, Marc Erhardt. ✉email: michael.kolbe@helmholtz-hzi.de; marc.erhardt@hu-berlin.de

The Gram-negative enteropathogen *Salmonella enterica* serovar Typhimurium (*Salmonella* Typhimurium) uses a variety of strategies to successfully enter and replicate within a host. In this respect, bacterial motility enables the directed movement of the bacteria towards nutrients or the target site of infection. A rotary nanomachine, the flagellum, mediates motility of many bacteria, including *Salmonella* Typhimurium[1]. Flagella also play a central role in other infection processes, involving biofilm formation, immune system modulation, and adhesion[2–5].

The eukaryotic plasma membrane plays an important role in the interaction of flagellated bacteria with host cells during the early stages of infection[6]. The flagella of *Salmonella* Typhimurium, *Escherichia coli* and *Pseudomonas aeruginosa* can function as adhesion molecules[7–9] mediating the contact to various lipidic plasma membrane components, including cholesterol, phospholipids, sulfolipids and the gangliosides GM1 and aGM1[10–13].

Structurally, the flagellum consists of three main parts: the basal body embedded within the inner and outer membranes of the bacterium, a flexible linking structure—the hook, and the long, external filament, which functions as the propeller of the motility device[14]. The filament is formed by more than 20,000 subunits of a single protein, flagellin[15]. Many *S. enterica* serovars express one of two distinct flagellins, FliC or FljB, in a process called flagellar phase variation[16]. FliC-expressing bacteria display a distinct motility behavior on host cell surfaces and a competitive advantage in colonization of the intestinal epithelia compared to FljB-expressing bacteria[17]. However, while the structure of FliC has been determined previously[18], the structure of FljB remained unknown.

The many thousand surface-exposed flagellin molecules are a prime target of the host's immune system. Accordingly, many flagellated bacteria have evolved mechanisms to prevent flagellin recognition, e.g. by posttranslational modifications of flagellin. Flagellin glycosylation is relatively common among Enterobacteriaceae[19], in *Campylobacter*, *Aeromonas*, and *Pseudomonas* species[20–22] and plays a critical role in adhesion, biofilm formation or mimicry of host cell surface glycans[23].

*Salmonella* Typhimurium does not posttranslationally glycosylate its flagellins. However, ε-N-methylation at lysine residues of flagellin via the methylase FliB has been reported[24–26]. Although flagellin methylation was first reported in 1959[25], the physiological role of the methylation remained elusive. Previous studies suggested that the absence of FliB had no significant effect on swimming and swarming motility[27,28]. However, the conservation of flagella methylation in *Salmonella* and other Enterobacteriaceae[19] suggests that methylated flagella are important for some other aspect of the life style of the bacteria. In the present study, we therefore investigated the hypothesis that flagella methylation contributes to some aspect of virulence of *Salmonella* Typhimurium.

We find that methylation of flagella facilitates adhesion of *Salmonella* Typhimurium to hydrophobic host cell surfaces. Thus, the posttranslational methylation of flagellin plays an important role for invasion of host cells, and accordingly, productive colonization of the host's epithelium.

## Results

### Methylated lysines residues in flagellin and structure of FljB.
Previous studies suggested that the flagellins of *Salmonella* Typhimurium are posttranslationally methylated, however, the identity of the methylated lysine residues remained largely unknown[25,26,29]. We performed mass spectrometry analyses with high sequence coverage of both flagellins FliC and FljB isolated from *Salmonella* Typhimurium genetically locked in expression of FliC (*fliC*^ON) or FljB (*fljB*^ON), respectively, and isogenic mutants of the methylase FliB (Δ*fliB*) (Supplementary Fig. 1). In order to map the identified ε-N-methyl-lysine residues to the structure of both flagellins, we determined the crystal structure of FljB (Fig. 1, Supplementary Note 1). The tertiary structure of FljB resembles, similar to FliC, a boomerang-shape with one arm formed by the D1 domain and the other formed by D2 and D3[18]. However, the variable D3 domain of FljB is rotated about 90° around the axis defined by the D2–D3 arm compared with FliC, resulting in the widening angle of about 20° between the two boomerang arms, consistent with the recently reported structure of flagellar filaments composed of FljB by cryo-electron microscopy[30] (Fig. 1b, c, Fig. 2a). Interestingly, the methylated lysine residues are primarily located in the surface-exposed D2 and D3 domains of both flagellins (Fig. 1d, Fig. 2a, Supplementary Note 2). Notably, except for two lysines in FliC and three in FljB, the detected lysine residues were methylated only in the presence of the methylase FliB and most of the lysines conserved between both flagellins were methylated (Fig. 1d, Supplementary Note 2).

We next aligned the amino acid sequences of FljB and FliC up- and downstream of the identified ε-N-methyl-lysine residues (± 6 residues, Supplementary Fig. 2). Although no clear consensus sequences could be determined, we found prevalence of small (Ala, Gly, Thr, Val, Ser) and negatively charged (Asp) residues around the methylated lysines. Interestingly, a scan of the local amino acid sequences that surround methylated lysines using ScanProsite[31] matched the profile of the bacterial Ig-like domain 1 (Big-1) for 10 modifications in both FljB and FliC, although with low confidence level (Supplementary Table 2). We note that the Big-1 domain is present in adhesion proteins of the intimin/invasin family, which are crucial in bacterial pathogenicity mediating host-cell invasion or adherence[32–34], suggesting a similar role for bacterial flagellins. Due to the absence of a clear amino acid consensus sequence, we next investigated the secondary structure elements around the methylation sites. We could not, however, identify a specific, conserved secondary structure element (Supplementary Fig. 3). We note that the low number of modified lysines in helices can be explained with the scarcity of this secondary structure element in the D2 and D3 domains.

### Flagella methylation contributes to host cell invasion.
We next investigated if methylation of FliC and FljB affects flagellar assembly and motility in *Salmonella* Typhimurium. The levels of non-methylated flagellin secreted from a Δ*fliB* mutant strain were comparable to secretion of methylated FliC or FljB (Fig. 3a). Immunostaining of flagella from the WT and a Δ*fliB* mutant strain revealed no significant differences in flagella assembly and flagella numbers per cell body (WT = 2.2 ± 1.8; Δ*fliB* = 2.2 ± 1.5) (Fig. 3b). In agreement with earlier reports[27,28], swimming motility of Δ*fliB* mutant strains in semi-solid agar plates was also not affected (Fig. 3c).

The absence of a motility phenotype in non-methylated flagellin mutants suggested that flagellin methylation might play a role in *Salmonella* virulence. We thus co-infected streptomycin-pre-treated mice[35] with the wildtype (WT) and an isogenic Δ*fliB* mutant (Fig. 2b). Organ burden analysis 2 days post-infection revealed that the Δ*fliB* strain was significantly outcompeted by the WT in the gastroenteritis mouse model, especially in the cecal tissue (Fig. 2b, competitive indices >1), suggesting that methylated flagella appear to enhance efficient colonization of the intestinal epithelium.

We next tested if methylated flagella contribute to efficient adhesion and invasion of epithelial cells in vitro (Fig. 4a). We first infected murine MODE-K epithelial cells with the WT and *Salmonella* Typhimurium strains deficient in the methylase FliB

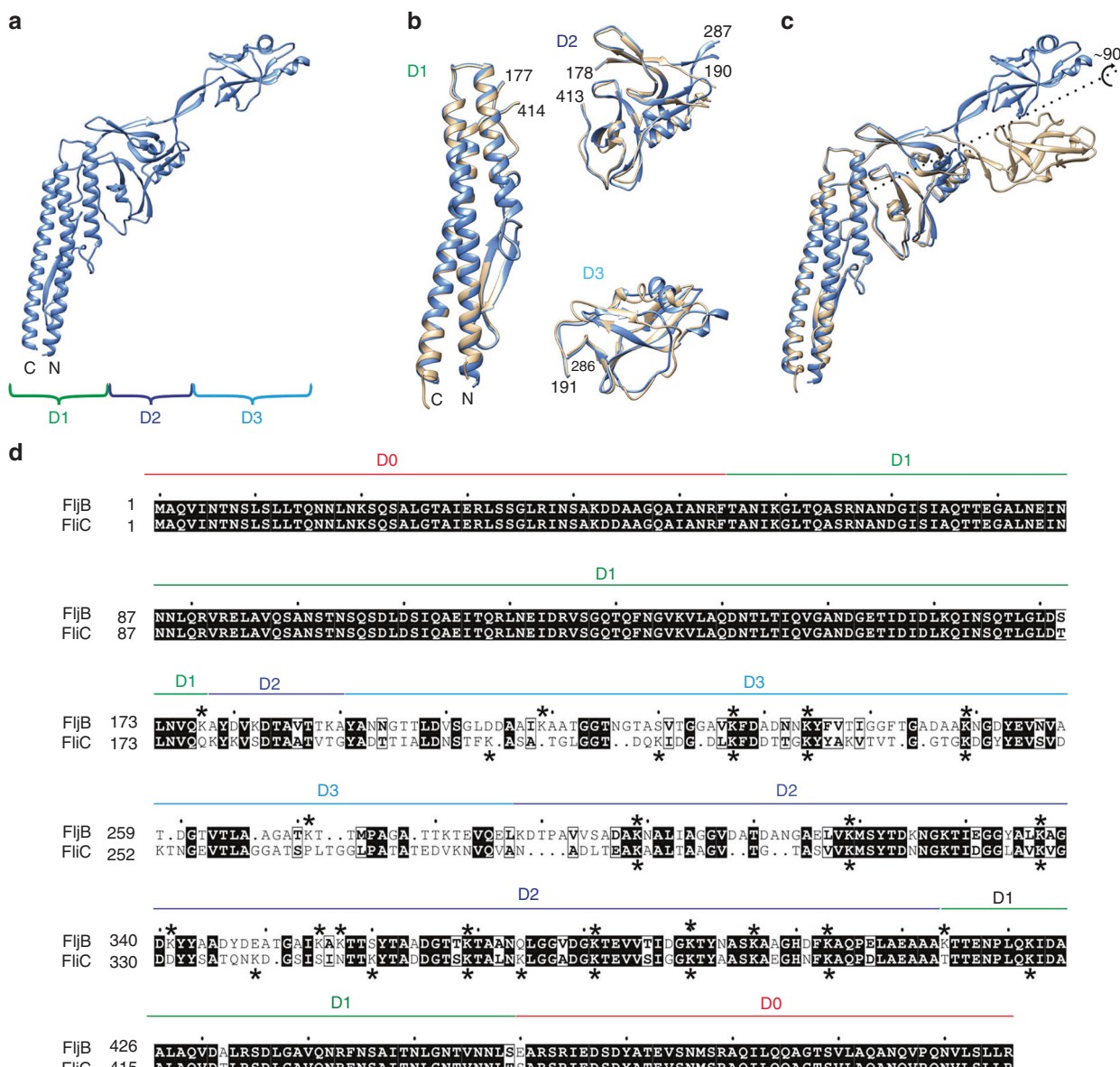

**Fig. 1 Structure of FljB and methylation pattern of the flagellins FliC and FljB. a** Cartoon representation of the structure of truncated FljB. The position of the N- and C-termini is shown at the end of the coiled-coil, and the extension of the domains D1, D2, and D3 is indicated below the structure. **b** Structural superposition of the individual domains. FljB domains are represented in blue, FliC in beige. The N- and C-termini of the structure are indicated, as well as the FljB numbering of the residues at the ends of the polypeptide segments defining the domains. **c** The structures of truncated FljB (blue) and truncated FliC (beige) have been superposed according to the D1 and D2 domains. The domain D3 shows a rotation of about 90° around an ideal axis starting from one end of the coiled-coil in D1 and passing through D2 (shown as dotted line). **d** Alignment of FljB and FliC and methylation pattern. Residues of domains D1, D2, and D3 have been aligned based on the structural superposition obtained with the EBI PDBeFold v2.59 server (51), either superposing the D1–D2 domains or the D3 domain. Residues of domain D0 have been aligned based on the sequence. The extent along the sequence of the domains is indicated above the alignment. Methylated lysine residues are indicated with stars above or below the alignment for FljB or FliC, respectively.

and determined the number of intracellular bacteria. Invasion was reduced about 50% for the Δ*fliB* mutant strain independently of the flagellin type (Fig. 4b, top). We also observed a similar invasion defect for the Δ*fliB* mutant when we forced contact of the bacteria with the epithelial cells using centrifugation (Fig. 4b, bottom), suggesting that the invasion defect of the Δ*fliB* mutant did not depend on active bacterial motility. We next confirmed that the observed invasion phenotype was due to the lack of *fliB* by complementing expression of *fliB* from an inducible P$_{tet}$ promoter at its native chromosomal locus. Addition of anhydrotetracycline (AnTc) induced *fliB* expression comparable to levels of the WT and restored invasion of MODE-K epithelial cells (Supplementary Fig. 4). We further tested if the observed invasion defect was dependent on the assembly of the methylated flagellar filament. A hook deletion mutant (Δ*flgE*) does not express flagellin, whereas a mutant of the hook-filament junction proteins (Δ*flgKL*) expresses and secretes flagellin, but does not assemble the flagellar filament. The methylase FliB is expressed in both Δ*flgE* and Δ*flgKL* mutant backgrounds[27]. We observed in neither the Δ*flgE* nor the Δ*flgKL* mutant a difference in MODE-K epithelial cell invasion in the presence or absence of FliB, suggesting that methylated flagellin must assemble into a functional flagellar filament in order to facilitate epithelial cell invasion (Supplementary Fig. 5).

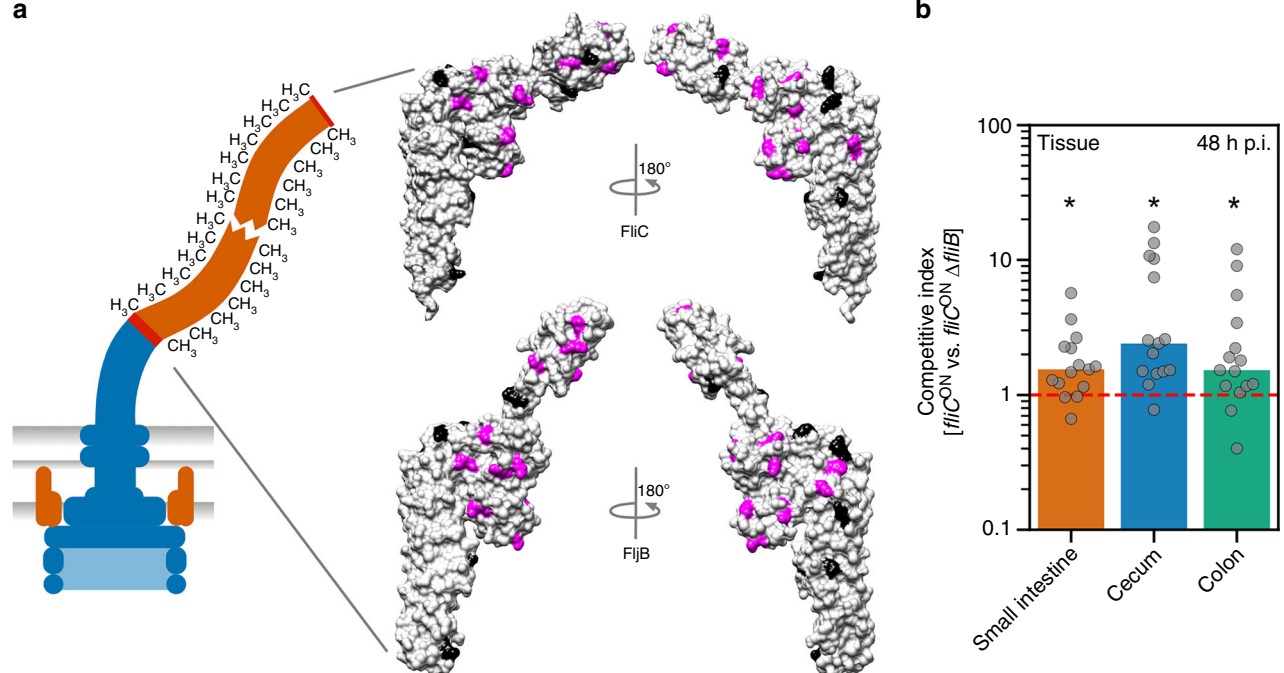

**Fig. 2 Surface-exposed methylation of flagellin contributes to efficient colonization of the murine intestine. a** Schematic of a methylated flagellar filament and surface representation of the structure of FliC (top) and FljB (bottom). Methylation sites are highlighted in magenta and non-methylated lysines in black. **b** Streptomycin pre-treated C57BL/6 mice were infected with $10^7$ CFU of the FliC-expressing WT ($fliC^{ON}$) and isogenic $\Delta fliB$ mutant, each harboring a different antibiotic resistant cassette. The organ burden (small intestine, colon and cecum, respectively) was determined 2 days post-infection and used to calculate the competitive indices (CI). Each mouse is shown as an individual data point and significant differences were analyzed by a two-tailed Wilcoxon Signed Rank test. The bar graph represents the median of the data and asterisks indicate a significantly different phenotype to the value 1 (* = $p < 0.05$. WT vs. $\Delta fliB$ small intestine: $p = 0.0135$, WT vs. $\Delta fliB$ cecum: $p = 0.010$, WT vs. $\Delta fliB$ colon: $p = 0.0384$). $n = 15$ biologically independent animals. Source data are provided as a Source Data file.

**Methylated flagella facilitate bacterial adhesion**. Our results presented above demonstrate that the presence of an assembled, methylated flagellar filament, but not the ability to move per se, contributes to the observed defect of *Salmonella* to invade epithelial cells. We note that our invasion assays report the number of bacteria that were successful in entering the eukaryotic cells. The assay thus reports two separate virulence mechanisms, i.e. successful invasion of epithelial cells is dependent on the *Salmonella* pathogenicity island-1 (*spi*-1) encoded injectisome and requires prior adhesion of the bacteria to the surface of eukaryotic cells, which is mediated through adhesion factors including pili, fimbriae and flagella.

Accordingly, we hypothesized that methylated flagella primarily facilitate bacterial adhesion to epithelial cells. We therefore investigated adhesion of *Salmonella* Typhimurium to MODE-K epithelial cells. In order to dissect flagella methylation-dependent adhesion from flagella methylation-dependent invasion of the epithelial cells, we employed *Salmonella* mutants deleted for *spi*-1, which renders the bacteria unable to invade epithelial cells in an injectisome-dependent manner. We found that adhesion of $\Delta spi$-1 *Salmonella* mutants to MODE-K epithelial cells was reduced up to 50% for strains deficient in flagellin methylation (Fig. 4c).

We next generated chromosomal substitution mutants of surface-exposed lysine residues in the D2 and D3 domains of FliC or FljB. As shown in Supplementary Fig. 6, the substitution of nine surface-exposed lysines with arginine in FliC (FliC-K9R) displayed an invasion defect similar to a $\Delta fliB$ mutant. We note that the FliC-K9R mutation decreased motility to 70% of the WT, which might contribute to the more pronounced invasion defect of FliC-K9R compared to the $\Delta fliB$ mutant. Replacing five surface-exposed lysine residues in the D3 domain of FliC with

arginine (FliC-K5R) did not affect motility, but affected bacterial surface adhesion as described below (Supplementary Fig. 6a, c). We further found that a complete removal of the D3 domains of FliC (FliCΔD3) or FljB (FljBΔD3) did not affect motility (Supplementary Fig. 6d). Deleting the D3 domain also removes the majority of methylated lysines and, interestingly, FliCΔD3 and FljBΔD3 strongly affected invasion of MODE-K epithelial cells (Supplementary Fig. 6e).

We further analyzed if the methylation-dependent invasion phenotype was eukaryotic cell-line specific (Supplementary Fig. 7). The human epithelial cell line E12[36] and murine intestinal epithelial cell line Cl11 mimic the native intestinal environment in vitro. Similar to the observed invasion rate of MODE-K cells, a $\Delta fliB$ mutant strain displayed a two-fold decreased invasion rate of the human and murine epithelial cell lines compared with the WT. Similarly, in murine epithelial-like RenCa cells, the invasion rate of a $\Delta fliB$ mutant was decreased. Invasion into the murine fibroblast cell lines NIH 3T3 and CT26, however, was independent of flagellin methylation, suggesting that the observed invasion phenotype is cell type-specific for epithelial-like cells.

**Methylated flagella promote adhesion to hydrophobic surfaces**. Our results described above demonstrate that methylated flagella promote bacterial adhesion to epithelial cells, e.g. through interaction with hydrophobic patches, surface-exposed proteinaceous receptors or glycostructures. However, we did not observe a significant flagella methylation-dependent effect on adhesion of *Salmonella* to various extracellular matrix proteins, nor to the oligosaccharide mannose, which has previously been shown to mediate adhesion of *Salmonella* and *E. coli* to eukaryotic cells

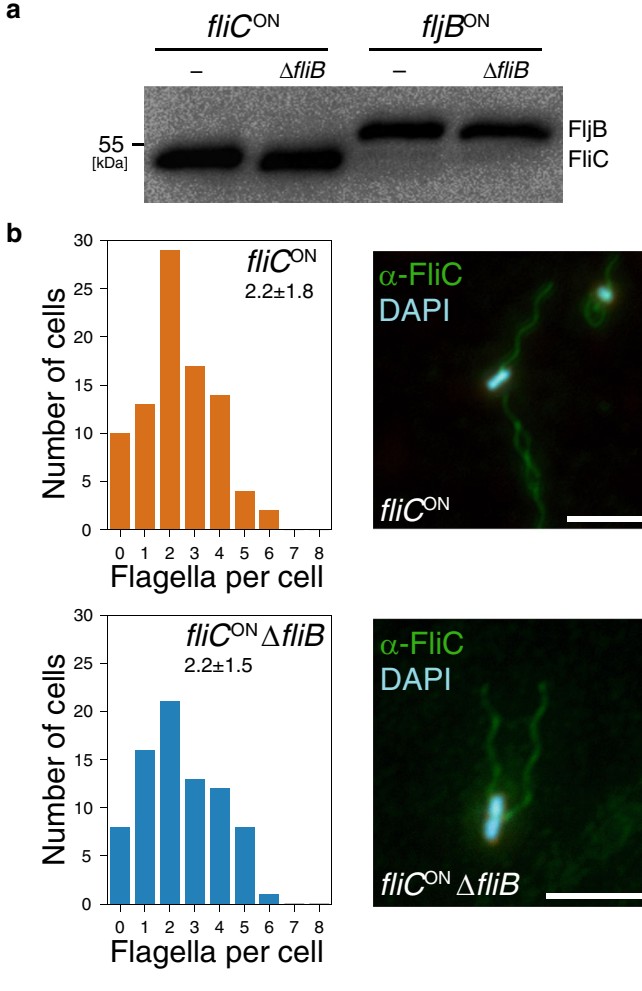

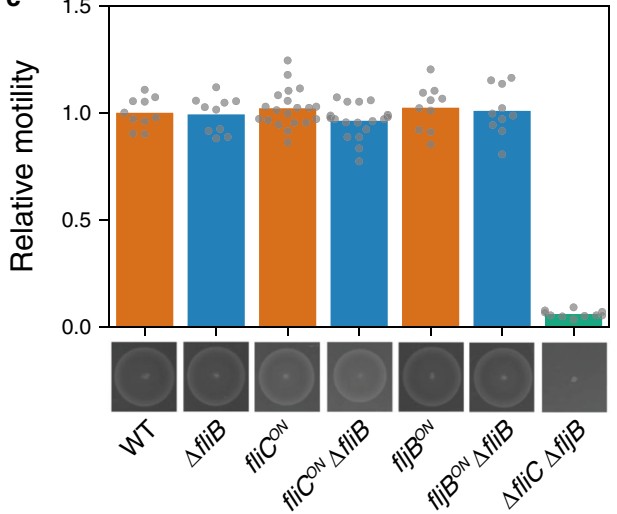

**Fig. 3 Effect of flagella methylation on swimming motility and flagellar assembly. a** Secreted flagellins from culture supernatants of *Salmonella* Typhimurium strains locked in expression of either FliC (*fliC*^ON) or FljB (*fljB*^ON). Secreted proteins were precipitated by addition of 10% TCA and fractionated according to their molecular weight by SDS-PAGE. Immunoblotting was performed using α-FliC/FljB antibodies (1:5,000). A representative immunoblot is shown. The experiment has been repeated three times with similar results. **b** Left: Histograms of the number of flagella per cell of the WT (*fliC*^ON) and an isogenic *fliB* mutant (*fliC*^ON Δ*fliB*). $n = 79$ bacteria for *fliC*^ON; $n = 89$ bacteria for *fliC*^ON Δ*fliB*. Average flagella numbers were calculated by Gaussian non-linear regression analysis. Right: Representative flagella immunostaining images. Flagellar filaments were immunostained using α-FliC primary (1:1000) and α-rabbit conjugated AlexaFluor 488 secondary antibodies (1:1000; green). DNA was stained with DAPI (blue). Scale bar = 5 μm. **c** Motility phenotypes of the WT and *fliB* mutants were analyzed in soft-agar plates containing 0.3% agar and quantified after 4 h incubation at 37 °C. Bottom: representative motility plate. Top: The diameters of the motility swarm were measured and normalized to the control strain. The bar graphs represent the mean of $n = 10$ biologically independent samples for the WT, Δ*fliB*, *fljB*^ON, *fljB*^ON Δ*fliB*, and Δ*fliC*Δ*fljB* and $n = 20$ biologically independent samples for *fliC*^ON and *fliC*^ON Δ*fliB*. Replicates are shown as individual data points. Source data are provided as a Source Data file.

We next established an in vitro assay to investigate the possibility that the increased hydrophobicity of methylated flagella promotes bacterial adhesion to the hydrophobic plasma membrane (Fig. 5c). This assay is based on the binding of *Salmonella* to giant unilamellar vesicles (GUV) consisting of phosphatidylcholine (PC), the most abundant phospholipid in animal tissues. Notably, we observed a reduction in bacterial adhesion to GUVs consisting of PC for *Salmonella* Typhimurium strains deficient in flagellin methylation, but not for the non-flagellated Δ*flgK* mutants. In support, a mutant replacing several surface-exposed lysine residues with arginine in the D3 domain of FliC (FliC-K5R) displayed reduced adhesion to GUVs consisting of PC (Supplementary Fig. 6c). In addition, non-motile, but flagellated bacteria (Δ*motAB*) were less adherent, which supports previous observations that actively rotating flagella are important for the initial interaction with surfaces before biofilm formation[40] (Fig. 5c).

In order to validate the contribution of flagella methylation on adhesion of *Salmonella* to phospholipids, we tested adhesion of the WT and Δ*fliB* mutants to phosphatidylglycerol vesicles. Phosphatidylglycerol (PG) is a minor lipid in higher eukaryotes[41] and a negatively charged molecule as opposed to zwitterionic PC. We speculated that Δ*fliB* mutants might display increased binding to GUVs consisting of PG due to exposed lysines in flagellin. As shown in Fig. 5d, we indeed observed significantly increased adhesion of Δ*fliB* mutants with non-methylated flagella compared to the respective WT strains.

## Discussion

Flagella-dependent motility is crucial for *Salmonella* pathogenesis by enabling directed movement towards host epithelial cells. However, flagella not only play a role in bacterial motility, but also in colonization, adhesion, and biofilm formation[40,42,43]. Concerning the role of flagella as an adhesion molecule, it is important to note that the flagellar filament is made of several thousand copies of a single protein, flagellin, which can mediate various interactions with surfaces.

Here, we describe a novel mechanism of flagella-dependent adhesion to surface-exposed hydrophobic molecules. This adhesion phenotype is facilitated by methylation of surface-exposed

using type I fimbriae[37–40] (Supplementary Fig. 8). We therefore reasoned that the addition of hydrophobic methyl groups to surface-exposed lysine residues (Fig. 2, Supplementary Fig. 9) might affect the hydrophobicity of the flagellar filament and through this mechanism promote bacterial adhesion to hydrophobic molecules present on the surface of epithelial cells. Consistently, the surface hydrophobicity (So) of purified FliC and FljB flagella was significantly reduced in the absence of lysine methylation (Fig. 5a, b).

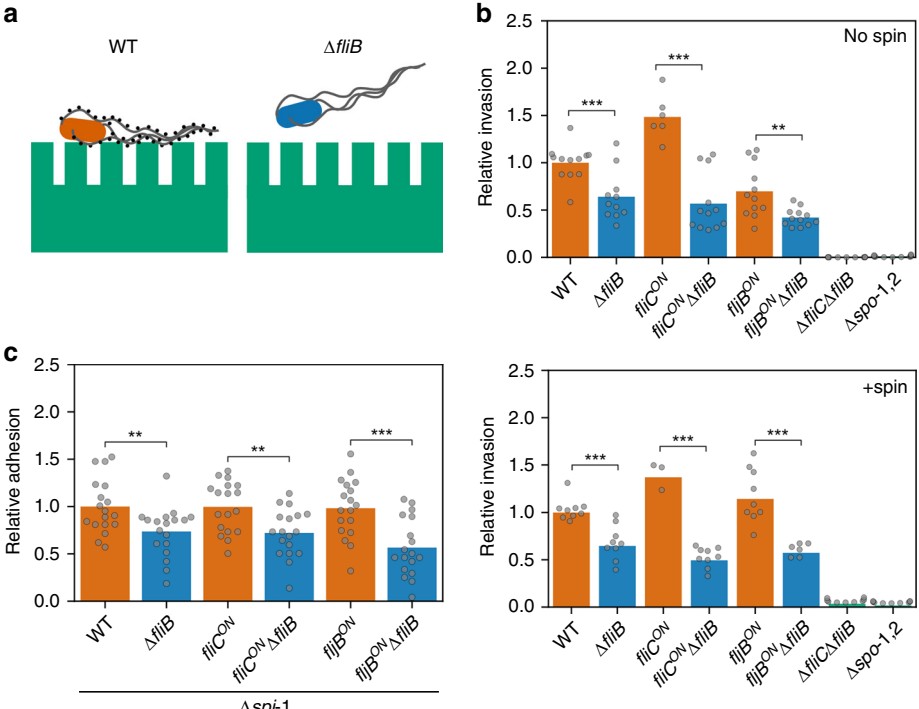

**Fig. 4 Flagella methylation facilitates eukaryotic cell adhesion and invasion. a** Schematic illustration of *Salmonella* Typhimurium adhesion to eukaryotic epithelial cells, which is facilitated by methylated flagella. **b** Invasion of MODE-K murine epithelial cells by *Salmonella* Typhimurium expressing methylated or non-methylated flagella. Reported are relative invasion rates of MODE-K epithelial cells for various flagellin methylation mutants without (top: no spin) or with forced contact of the bacteria by centrifugation (bottom: +spin). The bar graphs represent the mean of the reported data. Top: $n = 12$ biologically independent samples for the WT, *fliC*ON Δ*fliB*, *fljB*ON, *fljB*ON Δ*fliB*, Δ*fliC* Δ*fljB*, and Δ*spi*-1,2, $n = 11$ biologically independent samples for Δ*fliB* and $n = 6$ biologically independent samples for *fliC*ON. Replicates are shown as individual data points and statistical significances were determined by a two-tailed Student's *t* test (** = $p < 0.01$; *** = $p < 0.001$. WT vs. Δ*fliB*: $p = 0.0010$, *fliC*ON vs. *fliC*ON Δ*fliB*: $p < 0.0001$, *fljB*ON vs. *fljB*ONΔ*fliB*: $p = 0.0035$). Bottom: n = 9 biologically independent samples for the WT, Δ*fliB*, *fliC*ONΔ*fliB*, *fljB*ON, Δ*fliC*Δ*fljB* and Δ*spi*-1,2, $n = 3$ biologically independent samples for *fliC*ON and $n = 6$ biologically independent samples for *fljB*ONΔ*fliB*. Replicates are shown as individual data points and statistical significances were determined by a two-tailed Student's *t* test (*** = $p < 0.001$. WT vs. Δ*fliB*: $p = 0.0002$, *fliC*ON vs. *fliC*ONΔ*fliB*: $p < 0.0001$, *fljB*ON vs. *fljB*ONΔ*fliB*: $p = 0.0004$). **c** Adhesion of *Salmonella* Typhimurium to MODE-K epithelial cells is reduced in the absence of flagella methylation. Adhesion was monitored using *Salmonella* Typhimurium strains deleted for *spi*-1 in order to prevent invasion of the eukaryotic host cells. The bar graphs represent the mean of the reported relative invasion rate data normalized to the inoculum. $n = 18$ biologically independent samples. Replicates are shown as individual data points and statistical significances were determined by a two-tailed Student's *t* test (** = $p < 0.01$; *** = $p < 0.001$. WT vs. Δ*fliB*: $p = 0.0069$, *fliC*ON vs. *fliC*ONΔ*fliB*: $p = 0.0033$, *fljB*ON vs. *fljB*ONΔ*fliB*: $p = 0.0003$). Source data are provided as a Source Data file.

lysine residues of flagellin by the methylase FliB. Flagellin methylation was first described in *Salmonella* in 1959[24–26]; however, the physiological relevance remained elusive. We demonstrate that FliB-mediated flagellin methylation is crucial for *Salmonella* pathogenesis in the mouse model and contributes significantly to adhesion and thus invasion of epithelial cells in vitro, but does neither affect swimming motility nor flagella assembly (Fig. 3). Analysis of the surface hydrophobicity of purified flagella revealed that methylation of the filament subunits increases the hydrophobicity of the outer surface of the flagellar filament, while the lumen of the flagellar filament seems not to be affected (Fig. 5a, b, Supplementary Fig. 9). We note that the preferential methylation of surface-exposed lysine residues suggests that the methylation of lysines depends on some structural feature of flagellin and might occur only after flagellin has at least partially been folded. Further, we found that a single flagellin molecule can contain up to 16 (FliC) or 18 (FljB) surface-exposed methylation sites. Since a flagellar filament is made up of up to 20,000 flagellin copies, the methylation of flagellin subunits might substantially increase the overall hydrophobicity of the flagellum. Consistently, we found that adhesion to the surface of epithelial host cells and phosphatidylcholine vesicles was enhanced by methylated flagella. In support, flagella have been recently

implicated to mediate adhesion to abiotic surfaces through hydrophobic interactions[44,45]. We thus speculate that bacteria use flagella to explore the host cell surface as suggested previously[46] and actively rotating flagella might be able to penetrate the lipid bilayer and interact with the fatty acids buried inside the plasma membrane. Increasing the surface hydrophobicity of the flagellar filament through methylation might improve those hydrophobic interactions for productive adhesion to eukaryotic host cells.

Flagellin Methylation Islands (FMI) and thus modifications of flagellins by methylation are common in Enterobacteriaceae[19]. In addition to *Salmonella*, many bacterial species including *Yersinia*, *Enterobacter*, *Franconibacter*, and *Pantoea* contain chromosomal FMI loci, which encode orthologues of FliB. Interestingly, a mutant of a FliB homolog in *Aeromonas* also reduced the adherence to HEp2-cells[47]. In summary, FliB-dependent methylation of flagella might represent a general mechanism facilitating adhesion to hydrophobic host cell surfaces in a broad range of bacterial species.

## Methods

**Ethics statement.** All animal experiments were performed according to guidelines of the German Law for Animal Protection and with permission of the local ethics committee and the local authority LAVES (Niedersächsisches Landesamt für

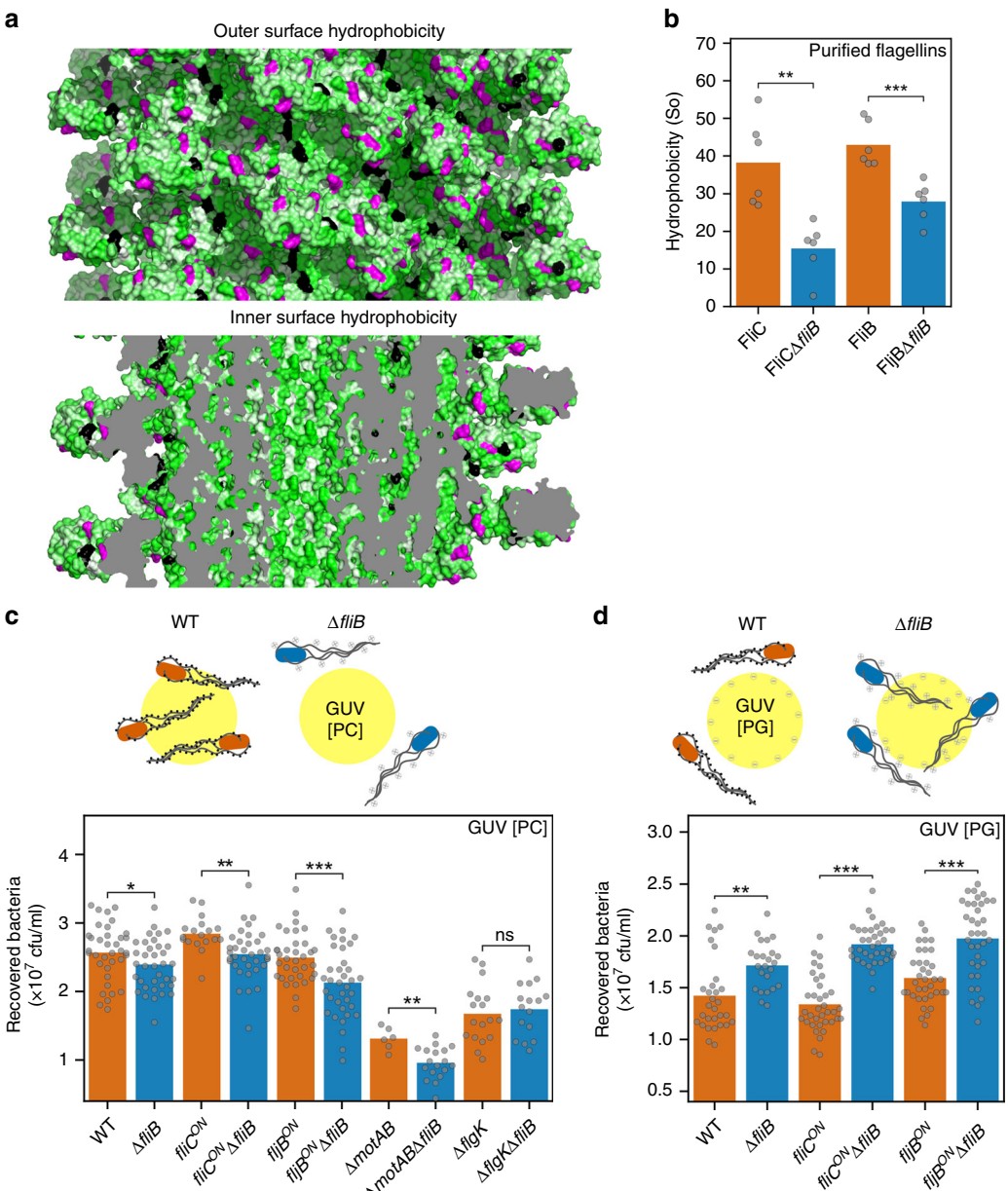

**Fig. 5 Flagella methylation mediates adhesion to hydrophobic surfaces. a** Methylation increases hydrophobicity of the flagellar filament outer surface. Surface hydrophobicity distribution of the outer (top) and inner (bottom) surface of the FliC flagellar filament[70] according to the Eisenberg scale[71] (from green to white indicates increasing hydrophobicity) with methylation sites highlighted in magenta and non-methylated lysines in black. **b** Measured surface hydrophobicity (So) of methylated and non-methylated (Δ*fliB*) flagellins using PRODAN on purified flagellar filaments. $n = 6$ independent experiments. Replicates are shown as individual data points and statistical significances were determined by a two-tailed Student's *t* test (** = $p < 0.01$; *** = $p < 0.001$. FliC vs. FliC Δ*fliB*: $p = 0.0020$, FljB vs. FljB Δ*fliB*: $p = 0.0009$. **c** Adhesion of *Salmonella* Typhimurium to giant unilamellar vesicles (GUV) consisting of phosphatidylcholine (PC) from egg chicken is facilitated by the presence of methylated flagella. Top: schematic illustration of the adhesion of *Salmonella* to PC-GUVs, which is facilitated by methylated flagella. Bottom: Quantified adhesion of *Salmonella* mutants to PC-GUVs. WT, Δ*fliB*, *fljB*^ON Δ*fliB*, *fliC*^ONΔ*fliB*: $n = 36$; *fljB*^ON: $n = 35$; *fliC*^ON, Δ*motAB* Δ*fliB*, Δ*flgK*: $n = 18$; Δ*flgK* Δ*fliB*: $n = 16$; Δ*motAB*: $n = 6$ biologically independent samples. Replicates are shown as individual data points and statistical significances were determined by a two-tailed Student's *t* test (* = $p < 0.05$; ** = $p < 0.01$; *** = $p < 0.001$; ns = not significant. WT vs. Δ*fliB*: $p = 0.0437$, *fliC*^ON vs. *fliC*^ONΔ*fliB*: $p = 0.0029$, *fljB*^ON vs. *fljB*^ONΔ*fliB*: $p = 0.0009$, Δ*motAB* vs. Δ*motAB* Δ*fliB*: $p = 0.025$). **d** Adhesion of *Salmonella* Typhimurium to GUVs consisting of phosphatidylglycerol (PG) is decreased by the presence of methylated flagella. Top: schematic illustration of the adhesion of *Salmonella* mutants to PG-GUVs. Bottom: Quantified adhesion of *Salmonella* mutants to PG-GUVs. The bar graphs represent the mean of the reported data. WT: $n = 30$; Δ*fliB*: $n = 24$; *fliC*^ON *fliC*^ONΔ*fliB*, *fljB*^ON and *fljB*^ONΔ*fliB*: $n = 36$ biologically independent samples. Replicates are shown as individual data points and statistical significances were determined by a two-tailed Student's *t* test (** = $p < 0.01$; *** = $p < 0.001$. WT vs. Δ*fliB*: $p = 0.0010$, *fliC*^ON vs. *fliC*^ONΔ*fliB*: $p < 0.0001$, *fljB*^ON vs. *fljB*^ONΔ*fliB*: $p < 0.0001$). The bar graphs represent the mean of the reported data. Source data are provided as a Source Data file.

Verbraucherschutz und Lebensmittelsicherheit) under permission number 33.19-42502-04-13/1191. Mice were housed in individually-ventilated cages with free access to autoclaved water and chow.

**Strains, media, and bacterial growth**. All bacterial strains used in this study are listed in Supplementary Table 3 and were derived from *Salmonella enterica* serovar Typhimurium strain SL1344. Bacteria were grown in lysogeny broth (LB) at 37 °C and growth was measured by optical density at 600 nm. For transductional crosses the generalized transducing phage P22 *HT105/1 int-201* was used[48]. Gene deletions or replacements were produced using λ-RED mediated homologous recombination[49]. All bacterial strains are available upon request.

**Cloning and purification of FljB for structural analysis**. The truncated gene *fljB* encoding for the protein residues 55-462 was amplified from *Salmonella* Typhimurium SL1344 by standard PCR method and cloned into the expression vector pET28a(+) using the restriction sites NheI and XhoI to generate a N-terminal His-tagged protein. The plasmid contained a A190V mutation as determined by sequencing. Standard protocols were performed for expression of His-tagged FljB$_{55-462}$ in BL21(DE3). The protein was purified from the soluble fraction using HisTrap HP and Superdex 75 columns (GE Healthcare) and eluted in the buffer 50 mM HEPES (pH 7.4), 150 mM NaCl.

**Crystallization and data collection**. FljB$_{55-462}$ was concentrated to 12–15 mg mL$^{-1}$ and crystals were grown at 18 °C by hanging drop vapour diffusion against 0.1 M Tris (pH 8.5), 20% (w/v) PEG4000, 24% (v/v) isopropanol. Diffraction data were collected using crystals flash-frozen in crystallization buffer. Measurements were carried out at the beamLine BL14.1 at the Helmholtz-Zentrum Berlin synchrotron Bessy II[50,51], using a wavelength of 0.918 Å and at 100 K, which allowed us to obtain a data set at 2.00 Å resolution. Crystals belong to space group *C*2, with one FljB molecule in the asymmetric unit (solvent content 51.6%). Indexing, integration, merging, and scaling were done using the program XDS[52].

**Crystal structure determination**. The structure was phased by molecular replacement with Phaser version 2.5.2[53], using the structure of the F41 fragment of FliC flagellin as search model (PDB 1IO1[18]). Cycles of manual building and refinement using Coot[54] and CNS Version 1.3[55] led to the final structure, which includes residues 55–459 of FljB with the mutation A190V and the residue S54 present in the crystallized construct. 299 water molecules were also placed. Structural comparison between FljB and FliC was performed using the server PDBeFold v2.59[56]. Molecular structure figures were generated using UCSF Chimera 1.13.1[57] (developed by the Resource for Biocomputing, Visualization, and Informatics at the University of California, San Francisco, with support from NIH P41-GM103311) and PyMol 2.2.3 (Schroedinger, LLC (2018). The amino acid alignment shown in Fig. 1d was generated using the server ESPript (http://espript.ibcp.fr)[58].

**Filament purification for mass spectrometry analysis**. Flagellar hook-basal-bodies with attached filaments of strains locked in either *fliC*$^{ON}$ or *fljB*$^{ON}$ and additionally harboring a Δ*fliB* mutation were purified as described[59] with minor modifications. Briefly, 500 mL logarithmically grown cultures were pelleted and re-suspended in ice-cold sucrose solution (0.5 M sucrose, 0.1 M Tris-HCl, pH 8). 3 mL 10 mg mL$^{-1}$ lysozyme and 3 mL 0.1 M EDTA were added drop wise into the samples and incubated at 4 °C for 30 min. Next, 3 mL 10% Triton X-100 and 3 mL 0.1 M MgSO4 were added followed by overnight incubation at 4 °C. After lysis, 3 mL 0.1 M EDTA, pH 11 was added and samples were centrifuged at 17,000 × *g* at 4 °C for 20 min. The supernatant was adjusted with 5 M NaOH to pH 11 and pelleted again. Hook-basal-bodies with attached filaments were subsequently collected by ultracentrifugation at 100,000 × *g* at 4 °C for 1 h. The pellet was re-suspended in buffer (10% sucrose, 0.1 M KCl, 0.1% Triton X-100, pH 11) and centrifuged at 17,000 × *g* at 4 °C for 20 min. Finally, purified flagellar filaments were washed in TET buffer (10 mM Tris-HCl, 5 mM EDTA pH 8, 0.1 % Triton X-100), air-dried and re-suspended in 200 μl TE buffer (10 mM Tris-HCl, 5 mM EDTA, pH 8). Samples were fractionated on NativePAGE 3–12% Bis–Tris gels (Invitrogen) and proteins were stained using Coomassie blue. Target bands were cut and prepared for mass spectrometry as detailed below.

**Mass spectrometry sample preparation**. FliC and FljB purified from the WT and a Δ*fliB* mutant were separated using SDS-PAGE. The corresponding bands were cut from the gel and each was cut into 1 × 1 mm pieces. An in-gel digestion was performed. In order to destain the gel pieces, a 50 mM ammonium bicarbonate (AmBiCa) in 50% acetonitrile (ACN) solution was added and incubated 30 min at room temperature to dehydrate the gel. After removal of the supernatant, 50 mM AmBiCa was added and incubated for 30 min at room temperature to rehydrate the pieces. This step was repeated two times. Disulfide bonds were reduced using 10 mM DTT in 50 mM AmBiCa for 30 min at 56 °C. After cooling to room temperature and removal of the supernatant, the reduced cysteines were alkylated using 55 mM iodoacetamide in 50 mM AmBiCa for 30 min at room temperature in the dark. After removal of the supernatant the gel pieces were dried in vacuo. Ice cold 50 mM AmBiCa in 10% ACN containing 12.5 ng μL$^{-1}$ trypsin was added and

digested overnight. Peptides were extracted by transferring the supernatant to a fresh collection tube and adding 50 mM AmBiCa in 10% ACN to the gel pieces and transferring the second supernatant into the same collection tube. Peptides were dried in vacuo and stored at −20 °C. Before measuring the peptides were reconstituted in 10 μL 0.1% formic acid (FA) and 1 μL was injected for measurement. All chemicals were purchased from Sigma-Aldrich. Trypsin was purchased from Promega.

**Mass spectrometry analysis**. Peptides were measured on a tandem mass spectrometer (Fusion, Thermo Fisher Scientific) coupled to a nano UPLC system (Ultimate 3000 RSLCnano, Thermo Fisher Scientific) with a nano-spray source. Peptides were trapped on a C18 reversed-phase trap column (2 cm × 75 μm ID; Acclaim PepMap trap column packed with 3 μm beads, Thermo Fisher Scientific) and separated on a 25 cm C18 reversed-phase analytical column (25 cm × 75 μm ID, Acclaim PepMap, 3 μm beads, Thermo Fisher Scientific). The column temperature was kept constant at 45 °C. Peptides were separated using a 2-step gradient starting with 3% buffer B (0.1% FA in ACN) and 97% buffer A (0.1% FA in H$_2$O) with a steady increase to 28% buffer B over 20 min and a second increase to 35% over 5 min with a subsequent ramping to 90% buffer B for 10 min followed by a 20 min equilibration to 3% buffer B at a constant flow rate of 300 nL min$^{-1}$. Eluting peptides were injected directly into the mass spectrometer. Data were acquired in positive ion mode using data dependent acquisition (DDA) with a precursor ion scan resolution of 1.2×10$^5$ at 200 $m·z^{-1}$ in a range of 300–1500 $m·z^{-1}$ with an automatic gain control (AGC) target of 2×10$^5$ and a maximum injection time of 50 ms. Peptides were selected for fragmentation using the 'TopSpeed' method with a threshold of 5000 intensity and a dynamic exclusion time of 30 s. Peptides were fragmented using higher-energy collision dissociation (HCD) in the C-Trap and fragment spectra were detected in the ion trap. Fragment spectra were recorded using the "Rapid" setting with a maximum injection time of 35 ms and an AGC target of 1×10$^4$ with the first mass set at 110 $m·z^{-1}$.

**Mass spectrometry data analysis**. Data were analyzed using the ProteomeDiscoverer 2.0 (Thermo Fisher Scientific) software. Spectra were identified using the Sequest HT search engine with precursor mass tolerance set to 10 ppm and the fragment mass tolerance set to 0.5 Da. Carbamidomethylation on cysteine was set as fixed modification and oxidation on methionine, acetylation on protein N-terminus as well as mono-, di- and tri-methylation on lysine were set as variable modifications. Trypsin was set as enzyme and three missed cleavages were allowed with a minimum peptide length of six amino acids. Spectra were searched against a *Salmonella* Typhimurium FASTA database obtained from UniProt in June 2016 containing 1821 entries and a contaminant database containing 298 entries. Sequence coverage maps were created using PatterLab for proteomics 4.0[60].

**Protein secretion assay**. Bacterial strains were grown over night in LB medium at 37 °C, diluted 1:100 into fresh medium and grown to mid-logarithmic phase. 1.5 mL samples of the culture supernatant were collected and precipitated by addition of 10% TCA and centrifuged for 1 h at 4 °C. The protein pellet was washed twice with acetone and air dried. Samples were adjusted to 200 OD units and fractionated under denaturing conditions using SDS-PAGE and immunoblotting was performed using primary α-FliC (Difco, catalog number 228241 *Salmonella* H Antiserum I. 1:5,000 in TBS-T) or α-FljB (Difco, catalog number 224741 *Salmonella* H Antiserum Single Factor 2, 1:5,000 in TBS-T) antibodies and detection was performed using secondary α-rabbit antibodies conjugated to horseradish peroxidase (Bio-Rad Immun-Star Goat Anti-Rabbit (GAR)-HRP Conjugate, catalog number 170-5046, 1:20,000 in TBS-T).

**Motility assay and immunostaining of flagellar filaments**. Swimming motility was analyzed in semi-solid agar plates containing 0.3% (w/v) agar. Single colonies were inoculated into the agar, and the plates were incubated at 37 °C for 4 h. Images were acquired by scanning the plates and the diameter of the swimming halos were determined using NIH ImageJ 1.48v and normalized to the WT control of the same plate. For immunostaining of flagellar filaments, logarithmically grown cells locked in expression of either FliC$^{ON}$ or FljB$^{ON}$ were fixed by addition of 2% formaldehyde and 0.2% glutaraldehyde for 10 min and immobilized on a poly-L-lysine coated coverslip. Flagellar filaments were immunostained using polyclonal α-FliC (Difco, catalog number 228241 *Salmonella* H Antiserum I, 1:1,000 in 2% BSA/PBS) or α-FljB (Difco, catalog number 224741 *Salmonella* H Antiserum Single Factor 2, 1:1,000 in 2% BSA/PBS) and secondary α-rabbit Alexa-Fluor488 (Invitrogen, catalog number A-11094, 1:1,000 in PBS). DNA was stained using DAPI (Sigma-Aldrich). Images were collected and processed as described before[28,61].

**Mouse infection studies**. Seven-week-old C57BL/6 mice (Janvier) were pretreated with 100 mg mL$^{-1}$ streptomycin. Mice were co-infected intragastrically with 10$^7$ colony forming units (CFU) each of two *Salmonella* Typhimurium strains that contained a different antibiotic resistance cassette. Small intestine, cecum and colon were isolated 2 days post-infection and plated on respective antibiotics resistance plates. The CFU were counted and reported as CFU per gram tissue. Competitive indices (CI) were calculated by normalizing the percentage of each strain to the inoculum and the challenge strain.

**Invasion and adhesion assays.** The murine epithelial cell lines MODE-K[62] and Cl11, the murine epithelial-like cell line Renca (CRL-2947), the human epithelial cell line HT29-MTX-E12 (E12)[36], the mouse fibroblast cell lines NIH-3T3 (CRL-1658) and CT26 (CRL-2638) were used for invasion assays. The immortalization and characterization of the muGob (Cl11) cells will be described elsewhere (Truschel et al., in preparation). Briefly, murine intestinal organoids were plated and infected with different lentiviruses encoding the CI-SCREEN gene library[63]. After transduction, the clonal cell line muGob (Cl11) was established, which has integrated the following recombinant genes of the CI-SCREEN library: Id1, Id2, Id3, Myc, Fos, E7, Core, Rex (Zfp42). The muGob (Cl11) cell line was cultivated on fibronectin/collagen-coated (InSCREENeX GmbH, Germany) well plates in a humidified atmosphere with 5% $CO_2$ at 37 °C in a defined muGob medium (InSCREENeX GmbH, Germany). $2.5 \times 10^5$ cells per mL were seeded in 24-well plates. For invasion of MODE-K epithelial cells, $5 \times 10^5$ cells per mL were seeded in 24-well plates and incubated overnight in a humidified atmosphere with 5% $CO_2$ at 37 °C. *Salmonella* strains were grown to mid-log phase, diluted in binding buffer (DMEM, 1% non-essential amino acids, 10% heat-inactivated FCS, 0.4% BSA, 20 mM HEPES pH 7.5) and added at a MOI of 10. The contact of the bacteria with the epithelial cells was forced by centrifugation for 5 min at $500 \times g$ and the infection was performed for 1 h in a humidified atmosphere with 5% $CO_2$ at 37 °C. After washing twice with 1× PBS, external bacteria were killed by addition of 100 µg·mL$^{-1}$ gentamicin for 1 h and cells were lysed using 1% Triton X-100. Serial dilutions of the lysate were plated to calculate the CFU per mL. All values were normalized to the control strain and invasion rates were calculated relative to the inoculum. For analysis of *Salmonella* Typhimurium adhesion to MODE-K epithelial cells, a modified assay was performed using *Salmonella* strains lacking *spi-1* to prevent injectisome-dependent invasion. Bacteria were added to MODE-K epithelial cells at a MOI of 10 and incubated for 1 h. Afterwards, the MODE-K cells were washed extensively to remove unbound bacteria and *Salmonella* CFU per mL were determined as described above without addition of gentamicin to kill external bacteria.

**RNA isolation and quantitative real-time PCR.** Strains were grown under agitating growth conditions in LB medium and total RNA isolation was performed using the RNeasy Mini kit (Qiagen). For removal of genomic DNA, RNA was treated with DNase using the TURBO DNA-free kit (Ambion). Reverse transcription and quantitative real-time PCRs (qRT-PCR) were performed using the SensiFast SYBR No-ROX One Step kit (Bioline) in a Rotor-Gene Q Lightcycler (Qiagen). Relative changes in mRNA levels were analyzed according to Pfaffl[64] and normalized against the transcription levels of the reference genes *gyrB*, *gmk* and *rpoD*[28].

**ECM adhesion assays.** For ECM protein adhesion assays, a 96-well plate pre-coated with a variety of ECM proteins was used (EMD Millipore; Collagen I, II, IV, Fibronectin, Laminin, Tenascin, Vitronectin). Wells were rehydrated according to the user's manual and $5\cdot10^7$ cells per mL were added. After incubation for 1 h at 37 °C, wells were washed extensively and 1% Triton X-100 was added. CFU per mL were calculated after plating of serial dilutions and normalized to the inoculum and a BSA control.

**Mannose binding assay.** Binding to mannose was determined as described before[65] with minor modifications. A black 96-well plate was coated with BSA or mannose-BSA (20 µg mL$^{-1}$ in 50 mM bicarbonate buffer pH 9.5) for 2 h at 37 °C, followed by blocking with BSA (10 mg mL$^{-1}$) for 1 h at 37 °C. Adjusted bacterial cultures (OD$_{600}$ 0.6) harboring the constitutive fluorescent plasmid pFU228-P$_{gapdh}$-mCherry were added to the wells to facilitate binding. After 1 h incubation at 37 °C, wells were washed with 1× PBS and fluorescence was measured using a Tecan plate reader (excitation 560 nm; emission 630 nm). Fluorescence relative to the binding to BSA was calculated from three technical replicates and the type-I fimbriae-inducible strain P$_{tet}$-*fimA-F* served as positive control.

**Filament isolation for surface hydrophobicity measurements.** Bacterial cultures were grown in LB media at 37 °C for 16 h in an orbital shaker incubator (Infors HT) at 80 rpm. Cells were harvest by centrifugation at $2000 \times g$ for 20 min. Cell pellets were re-suspended in TBS buffer pH 7.4 at 4 °C. The flagellar filaments were sheared-off using a magnetic stirrer at 500 rpm for 1 h, followed by centrifugation at $4000 \times g$ for 30 min. Supernatants were collected and ammonium sulfate was slowly added while stirring to achieve two-thirds saturation. After overnight incubation, the flagella were harvest by centrifugation at $15,000 \times g$ for 20 min and pellet was re-suspended in TBS buffer at pH 7.4. The quality of the purified flagella was checked by SDS-PAGE and transmission electron microscopy of negatively stained samples (microscope Talos L120C, Thermo Fisher Scientific).

**Hydrophobicity determination.** Protein surface hydrophobicity was measured according to a modification of the method of Kato and Nakai[66] using PRODAN[67]. A stock solution of 1 mM PRODAN (prepared in DMSO) was used, 8 µL was added to successive samples containing 1 mL of diluted flagella in 20 mM HEPES (pH 7.4), 150 mM NaCl. After homogenization by pipetting, the samples were incubated 10 min in the dark and the relative fluorescence intensity was measured.

All fluorescence measurements were made with a Cary Eclipse (Varian now Agilent) spectrofluorometer. Excitation and emission wavelengths were 365 nm and 465 nm, the slit widths were 5 and 5 nm. For standardization, BSA was used. Surface hydrophobicity (So) values were determined using at least duplicate analyses. Five measures per sample repeated three times were performed and the mean was used.

**Bacterial adhesion to liposomes.** Giant unilamellar vesicles (GUV) for PC (egg chicken (Sigma-Aldrich) or 1-palmitoyl-2-oleoyle-glycero-3-phosphocholine) and PG (1-palmitoyl-2-oleoyl-sn-glycero-3-phospho-(1′-rac-glycerol)) were prepared according to the polyvinyl alcohol (PVA)-assisted swelling method[68]. Gold-coated glass slides were obtained by thermal evaporation under vacuum (Evaporator Edwards model Auto 306, 0.01 nm s$^{-1}$, $(2-3) \times 10^{-6}$ mbar). A gold layer of $10 \pm 1$ nm was deposited on top of a chromium adhesion layer of $1 \pm 0.5$ nm. Prior to GUV formation in HEPES buffered saline solution (HEPES 20 mM pH 7.4, NaCl 150 mM), 1,2-distearoyl-sn-glycero-3-phosphoethanolamine-N-[PDP(polyethylene glycol)-2000] (DSPE-PEG-PDP) (Sigma-Aldrich) was mixed with L-α-phosphatidylcholine from egg chicken (Sigma-Aldrich) at a 3 % mass ratio that allows a direct covalent coupling of GUV onto gold surfaces. For the bacterial adhesion assay, a 5 µg mL$^{-1}$ GUV solution was deposited onto a gold-coated glass substrate and incubated 1 h for immobilization. Then, the surface was gently rinsed with buffer to remove non-immobilized liposomes. Subsequently bacterial culture at $10^8$ CFU per mL resuspended in HEPES buffer was carefully deposit on the surface and incubated for 1 h. Non-adherent bacteria were removed by washing with buffer. Finally, the adherent bacteria were detached by pipetting several times directly onto the immobilized liposomes with PBS pH 7.4. The collected samples were serially diluted and plated on LB agar for viable bacterial counts.

**Statistical analysis.** Data were analyzed using GraphPad Prism 6.0 h and statannot 0.1.0, a Python package for statistical annotations and analyses (https://github.com/webermarcolivier/statannot). Two-tailed Student's *t*-tests were used as appropriate. The *p* values < 0.05 were considered significant.

**Reporting summary.** Further information on research design is available in the Nature Research Reporting Summary linked to this article.

## Data availability
The authors declare that all data supporting the findings of this study are available within the paper and its supplementary information files. Full scans of the gels and blots are available in Supplementary Fig. 10. The source data underlying Figs. 2b, 3a–c, 4b, c, and 5b–d and Supplementary Figs. 4a–c, 5, 6a–e, 7a–e and 8a, b are provided as a Source Data file. Methylation data as well as corresponding mass spectrometry proteomics data have been deposited to the ProteomeXchange Consortium via the PRIDE[69] partner repository with the dataset identifier PXD017935 [https://doi.org/10.6019/PXD017935]. The coordinates of the flagellin FljB have been deposited in the worldwide Protein Data Bank (wwPDB) under accession number 6RGV [https://doi.org/10.2210/pdb6rgv/pdb].

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

## Acknowledgements

The authors thank Heidi Landmesser, Nadine Körner, Henri Galez and Pauline Adjadj for expert technical assistance, Juana de Diego and members of the Erhardt and Kolbe labs for useful discussions and for critical comments on the manuscript and Keiichi Namba for providing the atomic model of the FliC flagellar filament. We thank HZB for the allocation of synchrotron radiation beamtime and Uwe Müller for the support at the beamline BL14.1, Petra Dersch for kindly providing plasmid pFU228, Michael Hensel for providing P$_{tet}$-*fimA-F* mutant strains and Tobias May (InSCREENeX GmbH) for help in tissue culture and providing the epithelial-like cell line Cl11. J.A.H. acknowledges support by the President's Initiative and Networking Funds of the Helmholtz Association of German Research Centers (HGF) under contract number VH-GS-202. This work was supported in part by the Helmholtz Association Young Investigator grant VH-NG-932 and the People Programme (Marie Curie Actions) of the Europeans Unions' Seventh Framework Programme grant 334030 (to M.E.). The Helmholtz Institute for RNA-based Infection Research (HIRI) supported this work with a seed grant through funds from the Bavarian Ministry of Economic Affairs and Media, Energy and Technology (Grant allocation nos. 0703/68674/5/2017 and 0703/89374/3/2017) (to M.E. and T.S.). M.K., M.L. and C.W. were funded by the European Research Council under the European Community's Seventh Framework Programme under grant 311374 and through the President's Initiative and Networking Funds of the Helmholtz Association of German Research Centers (HGF). The HGF further supported T.S. by HGF impulse fund W2/W3-066. C.U. and J.H. acknowledge funding via Leibniz grant SAW-2014-HPI-4. The Heinrich-Pette-Institute, Leibniz Institute for Experimental Virology, is supported by the Freie und Hansestadt Hamburg and the Bundesministerium für Gesundheit (BMG). H.C. and L.L. acknowledge support by the French Ministry of Higher Education, Research and Innovation. Y.R., C.R., H.C. and L.L. acknowledge funding from the European Regional Development Fund ERDF and the Region of Picardy (CPER 2007–2020). We acknowledge support by the German Research Foundation (DFG) and the Open Access Publication Fund of Humboldt-Universität zu Berlin. The funders had no role in study design, data collection and analysis, decision to publish, or preparation of the manuscript.

## Author contributions

J.A.H., M.L., Y.R., M.K., and M.E. conceived the project, designed the study, wrote and revised the paper; J.A.H., M.L, H.C., L.L., M.K., J.H. and C.K. performed the experiments; J.A.H., M.L, H.C., J.H., C.K., C.U., G.A.G, Y.R., M.K., and M.E. analyzed the data; P.S., S.S., C.R., R.K.L., C.W., and K.T.H. contributed to experiments and performed strain construction; C.U., H.S., G.A.G., T.E.B.S., Y.R., M.K., and M.E. contributed funding and resources.

## Competing interests

The authors declare no competing interests.
