## [Peer Review File · Nature Communications]

Reviewers' comments:

Reviewer #1 (Remarks to the Author):

This is a very interesting and multi-disciplinary study that ascribes a function to flagellin methylation in *S. enterica*. Specifically, the authors determine the structures of the two flagellins, show which residues are methylated, and demonstrate that the enigmatic FliB methyltransferase is required for all but one of the methylations. Moreover, they show that FliB enhances *S. enterica* infections of mice, invasion of cell tissue culture, and adhesion to artificial vesicles.

The paragraph of the results became difficult to follow with a number of X out of Y lysine statements. I wasn't totally sure why I was being told all of these details about the individual lysines and how the comparisons nested with respect to one another. Could this paragraph be simplified somehow without the lysine counting?

Line 141. The reason to test virulence wasn't just the Big-1 domain, I presume, as line 91 suggested this was already hypothesized. Perhaps reassert as the previous hypothesis as the weak homology to Big-1 seemed a tenuous argument.

Line 225. Phosphatidylcholine is a neutral phospholipid as it has a positively charged head group that neutralizes the negative charge of the phosphate on an adjacent molecule. The adhesion of wild type and fliB mutants to phosphatidylglycerol vesicles should also be tested. PG is a negatively charged phospholipid and the fliB mutants might actually bind better (due to exposed lysines). This would provide a great control and help strengthen both the mechanism and the biological relevance as PC is, as stated, most abundant in animal membranes.

Reviewer #2 (Remarks to the Author):

In this article by Horstmann et al titled "Flagella methylation promotes bacterial adhesion and host cell invasion", the authors describe the functional effect of methylation of flagella in *Salmonella Typhimurium*. They posit that fliB methylates lysines on FliC and FliB which increases the hydrophobicity of flagella and results in increased adhesion to epithelial cells. This ultimately enables *Salmonella Typhimurium* to more efficiently colonize the gastrointestinal tract. The data presented is interesting and suggests that methylation of flagella contributes to the pathogenesis of *Salmonella*. The authors also present the structure of FliB which will be of interest to the scientific community. However, there are several major and minor comments outlined below which need to be addressed.

Major comments

(1) Invasion is dependent on the ability of bacteria to adhere to epithelial cells. Are the observed defects of non-methylated flagella in invasion due to the decreased ability to adhere to epithelial cells? What happens when you express invasion as a percent of adhered bacteria? It is not clear to this reviewer whether the authors are suggesting that methylation of flagella contributes to adherence and invasion (as 2 separate virulence mechanisms) or just adherence. Based on the data shown in Fig 3, it seems that the main effect is on adherence. As such, I would have liked to see the adherence data (Fig 3c- using a spi-1 mutant, but also using a strain with spi-1 intact) earlier. The invasion data is essentially irrelevant. This is highlighted by the fact that the authors don't even mention invasion in the abstract.

(2) Could fliB be mediating some other effect on flagella which is not related to methylation? It would have been nice to see some data showing the effect of reducing/demethylating flagella by another method. Did the authors try performing site directed mutagenesis to mutate some of the lysines (particularly ones that would be surface exposed)?

(3) The authors present data that certainly suggests that flagellin methylation contributes to Salmonella adhesion/invasion of gut epithelia; however, their conclusions that these processes are dependent upon methyl groups are too strong and not supported by their data. Firstly, the absence of fliB in FliC/FliB genetically-locked mutants diminishes adhesion to MODE-K cells and GUVs by 25% and at most 20%, respectively. Secondly, across three cell lines, fliCON Δ fliB mutants demonstrate at most a 50% decline in invasion whereas deletion of both flagellar types or SPI-1/2 completely abrogates invasion. Please tone down the language. E.g., methylation contributes to adherence.

(4) fliCON Salmonella slightly outcompete fliCON Δ fliB mutants within gut tissue (only a 1.5-2-fold difference). Demonstrating the importance of flagellin methylation in vivo is crucial to support the authors' conclusions. What happens when you infect mice with the strains separately? If methylation facilitates adherence in vivo, then it would have been nice to see some microscopy demonstrating colonization. Again, the conclusion on line 148-149 stating that "methylated flagella play an important role for efficient colonization of the intestinal epithelium" is overstated. At most, the data suggests that methylation appears to enhance colonization.

(5) The hypothesis that FliB-dependent methylation may be a general mechanism for adhesion to hydrophobic surfaces is certainly intriguing. As it stands, there is no data in the manuscript to suggest that this is conserved across Enterobacteriaceae; hence, the current title is too broad. Additional in vitro adhesion/invasion experiments with fliB-deficient strains from other genera would be required to support the title. This theoretically is feasible and would strengthen the manuscript tremendously. Alternatively, the title should be modified to reflect that the manuscript is focused on *S. Typhimurium*. E.g., "Flagella methylation promotes Salmonella Typhimurium adhesion and host cell invasion."

Minor

- 1) Why include results pertaining to the putative bacterial Ig-like domain if the confidence level from the ScanProsite analysis was within the "twilight zone". What conclusions can be drawn from this?
- 2) Adherence was measured on epithelial cells followed by phosphatidylcholine liposomes. The conclusion that methylated flagella promote adhesion to hydrophobic patches on epithelial cells cannot be drawn as this was never directly tested (lines 207-209). Please remove this statement.
- 3) Line 89-91. How does the lack of effect on motility suggest that methylation of flagella might be required for virulence?
- 4) Line 92-96. Please end the introduction with your hypothesis/goal/aims. Please remove the summary. The goal of the study was not clear.
- 5) Line 127. Please change "significances" to "significant differences".
- 6) Throughout the results section, there are a lot of concluding statements in which data has been over-interpreted/overstated instead of simply presented. E.g., Line 161. Please change sentence from "Invasion of MODE-K murine epithelial cells depends on methylated flagella" to "Invasion of MODE-K murine epithelial cells by Salmonella Typhimurium expressing methylated or non-methylated flagella."
- 7) Please add parentheses around "So" on line 198.
- 8) The Methods section needs proofreading for grammar.
- 9) Line 381. What was the source of the antibodies?
- 10) Line 414-416. What were the infection details for the adhesion assay? MOI of 10:1 for 1 hr?

Reviewer #3 (Remarks to the Author):

This is an interesting paper that reports the role of flagellin methylation, which has been obscured for a long time. The authors show that increase of hydrophobicity of the flagellar filament by methylation enhances adhesion to host cell surfaces, and thereby bacteria efficiently invade their

host cells. I think the results in this paper will attract broad readership and recommend this work for publication.

I have several comments that should be considered before publication.

1) The authors solved the structure of FljB, but I do not feel that the 3D structural information is utilized well in this paper. The distribution of the methylated lysine residues is shown (line 112-119, Fig1 and Fig3a), but that of the non-methylated ones is not. Non-methylated lysine residues also should be mapped on the FliC and FljB structures.

2) The authors analyzed characteristic features around the methylation sites only from the amino acid sequence level (line131-135 and Supplementary text 3). They should analyze and discuss them using the 3D structures.

3) The authors identified the methylation sites, but not mentioned about their methylation ratio. Are they methylated 100%? Do all the sites show the same methylation ratio? Some data or description may be needed.

4) The authors suggest that flagellin assemblies are formed in the cytoplasm (line258-259, and Supplementary text 3 line 111-114.), but this is unlikely because flagellin forms a complex with FliS in bacterial cytosol. FliS may prevent FliC from methylation. The complex structure is available from PDB (5MAW and 6GOW). Although the structures are from Bacillus, they may provide useful information.

5) Purification of FliC and FljB for mass spectroscopy was not described in method and should be shown.

Other minor comments

line 198

"the surface hydrophobicity S_o " -> "the surface hydrophobicity (S_o)"

line245

The sentence is weird.

line253

"Supplementary text S3" -> "Supplementary text S2 and FigS4"

line 309 and Table S1

"C2" "C" should be italic.

Reviewers' comments:

Reviewer #1 (Remarks to the Author):

This is a very interesting and multi-disciplinary study that ascribes a function to flagellin methylation in *S. enterica*. Specifically, the authors determine the structures of the two flagellins, show which residues are methylated, and demonstrate that the enigmatic FliB methyltransferase is required for all but one of the methylations. Moreover, they show that FliB enhances *S. enterica* infections of mice, invasion of cell tissue culture, and adhesion to artificial vesicles.

RE: We thank this reviewer for the detailed positive assessment and valuable comments. We note that we have carefully re-analyzed our mass spectrometry data of flagella isolated from the WT and the fliB mutant. We found that the one FliB-independently methylated lysine, which we reported in the previous version of our manuscript, is likely a false positive. We thus conclude that all detected, methylated lysines are dependent on the methylase FliB.

The paragraph of the results became difficult to follow with a number of X out of Y lysine statements. I wasn't totally sure why I was being told all of these details about the individual lysines and how the comparisons nested with respect to one another. Could this paragraph be simplified somehow without the lysine counting?

RE: Thank you for this suggestion. We have revised the visualization of methylated lysine residues on the flagellin structures (Fig. 1a) and moved details about the individual lysines into the legend of Fig. S2.

Line 141. The reason to test virulence wasn't just the Big-1 domain, I presume, as line 91 suggested this was already hypothesized. Perhaps reassert as the previous hypothesis as the weak homology to Big-1 seemed a tenuous argument.

RE: We agree and we have modified the sentence. It now reads: 'As mentioned above, the absence of a motility phenotype in non-methylated flagellin mutants (Supplementary Fig. S4, Supplementary Text S2) suggested that flagellin methylation might play a role in Salmonella virulence.'

Line 225. Phosphatidylcholine is a neutral phospholipid as it has a positively charged head group that neutralizes the negative charge of the phosphate on an adjacent molecule. The adhesion of wild type and fliB mutants to phosphatidylglycerol vesicles should also be tested. PG is a negatively charged phospholipid and the fliB mutants might actually bind better (due to exposed lysines). This would provide a great control and help strengthen both the mechanism and the biological relevance as PC is, as stated, most abundant in animal membranes.

RE: We thank this reviewer for this very interesting suggestion. We have tested adhesion of the WT and Δ fliB mutants to phosphatidylglycerol vesicles and we indeed observed significantly increased adhesion of Δ fliB mutants with non-methylated flagella compared to the respective WT strains (Fig. 3e).

Reviewer #2 (Remarks to the Author):

In this article by Horstmann et al titled "Flagella methylation promotes bacterial adhesion and host cell invasion", the authors describe the functional effect of methylation of flagella in *Salmonella Typhimurium*. They posit that *fliB* methylates lysines on *FliC* and *FliB* which increases the hydrophobicity of flagella and results in increased adhesion to epithelial cells. This ultimately enables *Salmonella Typhimurium* to more efficiently colonize the gastrointestinal tract. The data presented is interesting and suggests that methylation of flagella contributes to the pathogenesis of *Salmonella*. The authors also present the structure of *FliB* which will be of interest to the scientific community. However, there are several major and minor comments outlined below which need to be addressed.

RE: We thank the reviewer for the constructive and positive assessment of our work and we are grateful for the time and effort put into the review.

Major comments

(1) Invasion is dependent on the ability of bacteria to adhere to epithelial cells. Are the observed defects of non-methylated flagella in invasion due to the decreased ability to adhere to epithelial cells?

RE: Yes, the flagellum is an important adhesion factor and, in our experience, needed for efficient invasion of epithelial cells in vitro. Motility, i.e. the ability to actively swim towards epithelial cells, has a minor contribution and does not play a role in our invasion assays as we additionally forced contact of bacteria by centrifugation. This, together with the absence of a motility defect of the $\Delta fliB$ mutant, let us to conclude that the observed invasion defects of non-methylated flagella were primarily due to the decreased surface adhesion of $\Delta fliB$ mutant bacteria as demonstrated in Fig. 3.

What happens when you express invasion as a percent of adhered bacteria?

RE: The way how our invasion assays were performed does, unfortunately, not allow us to express invasion as a percent of adhered bacteria. All non-invading bacteria are killed by addition of gentamycin. We therefore performed a separate experiment using non-invading *spi-1* mutants in order to analyse bacterial adhesion to the epithelial cells. As now shown in Fig. 2c, we found that the $\Delta spi-1 \Delta fliB$ double mutant had a decreased ability to adhere to the epithelial cells compared to the $\Delta spi-1$ mutants with methylated flagella. Together with the absence of a motility defect, we thus concluded that our invasion assay is a useful assay to analyze defects in adhesion of non-methylated flagella mutants.

It is not clear to this reviewer whether the authors are suggesting that methylation of flagella contributes to adherence and invasion (as 2 separate virulence mechanisms) or just adherence. Based on the data shown in Fig 3, it seems that the main effect is on adherence. As such, I would have liked to see the adherence data (Fig 3c- using a *spi-1* mutant, but also using a strain with *spi-1* intact) earlier. The invasion data is essentially irrelevant. This is highlighted by the fact that the authors don't even mention invasion in the abstract.

RE: We agree that the primary effect of flagella methylation appears to increase the ability of the bacteria to adhere to surfaces e.g. epithelial cells. We have emphasized the fact that invasion and adhesion are two separate virulence mechanisms in the text and also moved the data showing adhesion of non-invasive spi-1 mutants to Fig. 2 as suggested.

(2) Could fliB be mediating some other effect on flagella which is not related to methylation?

RE: We did not observe any effect of the Δ fliB mutant on flagellin protein expression, flagellation or motility as shown in Fig. S5. We therefore concluded that any observed adhesion/invasion defect of the Δ fliB mutant was due to the absence of methylated, assembled flagellar filaments.

It would have been nice to see some data showing the effect of reducing/demethylating flagella by another method.

Did the authors try performing site directed mutagenesis to mutate some of the lysines (particularly ones that would be surface exposed)?

RE: Thank you for this suggestion. We generated chromosomal substitution mutants of surface-exposed lysine residues in the D2 and D3 domains of FliC. As shown in Fig. S8, the substitution of nine surface-exposed lysines with arginine in FliC (FliC_K9R) displayed an invasion defect similar to a Δ fliB mutant. We note that the FliC_K9R mutation decreased motility to 70% of the WT, which might explain the more pronounced invasion defect of FliC_K9R compared to the Δ fliB mutant.

In support, a mutant replacing several surface-exposed lysine residues with arginine in the D3 domain of FliC displayed reduced adhesion to GUVs consisting of phosphatidylcholine (Fig. S8c).

We further found that a complete removal of the D3 domains of FliC or FljB did not affect motility (Fig. S8d). Deleting the D3 domain also removes the majority of methylated lysines. Interestingly, deletion of FliC_ Δ D3 and FljB_ Δ D3 strongly affected invasion of epithelial cells (Fig. S8e).

(3) The authors present data that certainly suggests that flagellin methylation contributes to Salmonella adhesion/invasion of gut epithelia; however, their conclusions that these processes are dependent upon methyl groups are too strong and not supported by their data. Firstly, the absence of fliB in FliC/FljB genetically-locked mutants diminishes adhesion to MODE-K cells and GUVs by 25% and at most 20%, respectively. Secondly, across three cell lines, fliCON Δ fliB mutants demonstrate at most a 50% decline in invasion whereas deletion of both flagellar types or SPI-1/2 completely abrogates invasion. Please tone down the language. E.g., methylation contributes to adherence.

RE: Thank you for these suggestions. It was not our intention to claim that adhesion of *Salmonella* completely depends on methylation of its flagellar filament. It is clear that the Δ fliB mutant is still able to adhere to surfaces and invade epithelial cells, albeit with significantly decreased efficiency. Accordingly, we have toned down our language throughout the manuscript. In

particular, we state in the abstract that ‘...methylation of flagellin promoted invasion of epithelial cells...’ and ‘...enhanced flagella-dependent adhesion...’, and we modified the description and legend of Fig. 2 as suggested.

(4) fliCON Salmonella slightly outcompete fliCON Δ fliB mutants within gut tissue (only a 1.5-2-fold difference). Demonstrating the importance of flagellin methylation in vivo is crucial to support the authors’ conclusions. What happens when you infect mice with the strains separately?

RE: We have not performed mice infection experiments with single mutants only for the following reasons. It has been shown previously that in the gastroenteritis mouse infection model, motility is not essential for colonization of the intestine in single strain infections (Stecher, B. et al. *Infect Immun* 72, 4138-4150 (2004)). Only in competitive infections, the WT outcompeted a non-motile mutant. We thus concluded that large biological variation in single strain infections would mask the rather small, albeit significant differences observed in competition experiments.

If methylation facilitates adherence in vivo, then it would have been nice to see some microscopy demonstrating colonization. Again, the conclusion on line 148-149 stating that “methylated flagella play an important role for efficient colonization of the intestinal epithelium” is overstated. At most, the data suggests that methylation appears to enhance colonization.

RE: We have toned-down our language as suggested.

(5) The hypothesis that FliB-dependent methylation may be a general mechanism for adhesion to hydrophobic surfaces is certainly intriguing. As it stands, there is no data in the manuscript to suggest that this is conserved across Enterobacteriaceae; hence, the current title is too broad. Additional in vitro adhesion/invasion experiments with fliB-deficient strains from other genera would be required to support the title. This theoretically is feasible and would strengthen the manuscript tremendously. Alternatively, the title should be modified to reflect that the manuscript is focused on *S. Typhimurium*. E.g., “Flagella methylation promotes *Salmonella Typhimurium* adhesion and host cell invasion.”

RE: We agree that it is an intriguing possibility to speculate that flagellin methylation (or other posttranslational flagellin modifications) contribute to surface adhesion in other bacteria as well. However, analyzing the contribution of flagella methylation in virulence and surface adhesion of other bacterial species is out of scope of the present work. We have, however, submitted a grant proposal to study these possibilities in the future. Accordingly, we have changed the title to ‘Flagella methylation promotes adhesion and host cell invasion of *Salmonella Typhimurium*’ as suggested.

Minor

1) Why include results pertaining to the putative bacterial Ig-like domain if the confidence level from the ScanProsite analysis was within the “twilight zone”. What conclusions can be drawn from this?

RE: Although the low confidence level does not allow to draw conclusion about the functional relevance of the methylated sites, it is striking that the Big-1 profile is matched for most of them. The weak homology to a functional domain known to be involved in adhesion processes supports our observation that methylated flagella contribute to surface adhesion of *Salmonella*.

2) Adherence was measured on epithelial cells followed by phosphatidylcholine liposomes. The conclusion that methylated flagella promote adhesion to hydrophobic patches on epithelial cells cannot be drawn as this was never directly tested (lines 207-209). Please remove this statement.

RE: This section of manuscript has been extensively modified. We have toned-down our language in the mentioned sentence in order to suggest different possible modes of how methylated flagella enhance adhesion of bacteria to epithelial cells. The sentence now reads: ‘Our results described above demonstrate that methylated flagella promote bacterial adhesion to epithelial cells, e.g. through interaction with the hydrophobic patches, surface-exposed proteinaceous receptors or glycostructures.’

3) Line 89-91. How does the lack of effect on motility suggest that methylation of flagella might be required for virulence?

RE: No effect on motility, but conservation of methylation in *Salmonella* and other Enterobacteriaceae suggests that methylated flagella are important for some other aspect of the life style of the bacteria e.g. methylated flagella contribute to a virulence mechanism. We have simplified this sentence.

4) Line 92-96. Please end the introduction with your hypothesis/goal/aims. Please remove the summary. The goal of the study was not clear.

RE: Changed as suggested. The introduction now ends with our hypothesis that methylated flagella contribute to some aspect of the virulence of *Salmonella*.

5) Line 127. Please change “significances” to “significant differences”.

RE: Changed as suggested.

6) Throughout the results section, there are a lot of concluding statements in which data has been over-interpreted/overstated instead of simply presented. E.g., Line 161. Please change sentence from “Invasion of MODE-K murine epithelial cells depends on methylated flagella” to “Invasion of MODE-K murine epithelial cells by *Salmonella* Typhimurium expressing methylated or non-methylated flagella.”

RE: Changed as suggested. As noted above, we made the effort to tone-down our conclusions throughout the manuscript.

7) Please add parentheses around “So” on line 198.

RE: Changed and thank for spotting this.

8) The Methods section needs proofreading for grammar.

RE: We have proof-read and updated the Methods section.

9) Line 381. What was the source of the antibodies?

RE: The antibodies sources have been added: α -FliC (Difco 228241 *Salmonella* H Antiserum I), α -FliB (Difco 224741 *Salmonella* H Antiserum Single Factor 2); α -rabbit (Bio-Rad, Goat Anti-Rabbit conjugated to horseradish peroxidase).

10) Line 414-416. What were the infection details for the adhesion assay? MOI of 10:1 for 1 hr?

RE: Yes, *Salmonella* Δ *spi-1* strains were added at a MOI of 10 and incubated for 1h. The Method description was updated.

Reviewer #3 (Remarks to the Author):

This is an interesting paper that reports the role of flagellin methylation, which has been obscured for a long time. The authors show that increase of hydrophobicity of the flagellar filament by methylation enhances adhesion to host cell surfaces, and thereby bacteria efficiently invade their host cells. I think the results in this paper will attract broad readership and recommend this work for publication.

RE: We sincerely thank the reviewer for the time and effort put into the positive review of our manuscript.

I have several comments that should be considered before publication.

1) The authors solved the structure of FljB, but I do not feel that the 3D structural information is utilized well in this paper. The distribution of the methylated lysine residues is shown (line 112-119, Fig1 and Fig3a), but that of the non-methylated ones is not. Non-methylated lysine residues also should be mapped on the FliC and FljB structures.

RE: Thank you for this suggestion. We have updated the visualization of the methylated and non-methylated lysine residues shown in Fig. 1 and Fig. 3.

2) The authors analyzed characteristic features around the methylation sites only from the amino acid sequence level (line131-135 and Supplementary text 3). They should analyze and discuss them using the 3D structures.

RE: We tried to identify any characteristic structural features of the methylation sites as suggested. However, we were not able not identify a specific, conserved feature in respect to secondary structure elements around the methylation sites of FljB as summarized in the new Supplemental Fig. S4.

3) The authors identified the methylation sites, but not mentioned about their methylation ratio. Are they methylated 100%? Do all the sites show the same methylation ratio? Some data or description may be needed.

RE: Unfortunately, a quantification of methylated versus non-methylated peptides in a given sample is impossible due to different ionization efficiency, chromatographic profiles and higher frequency of missed cleavages in methylated peptides. Hence, very long peptides would be quantified against much shorter peptides, which have superior detection probability. We can only say that the lysines found occur also in a methylated form, which is highly dependent on FliB. For quantification and also co-occurrence of methylation sites top-down analysis would be necessary. This is however beyond the scope of this study.

Nevertheless, it is possible to qualitatively compare wildtype and mutant samples. Most methylation sites are lost entirely in the $\Delta fliB$ mutant, all others are greatly reduced.

During this re-analysis of our mass spectrometry data, we found that the one FliB-independently methylated lysine, which we reported in the previous version of our manuscript, is likely a false positive. We thus conclude that all

detected, methylated lysines are dependent on the methylase FliB.

4) The authors suggest that flagellin assemblies are formed in the cytoplasm (line258-259, and Supplementary text 3 line 111-114.), but this is unlikely because flagellin forms a complex with FliS in bacterial cytosol. FliS may prevent FliC from methylation. The complex structure is available from PDB (5MAW and 6GOW). Although the structures are from Bacillus, they may provide useful information.

RE: Thank you for spotting this. This statement was poor wording on our side. We agree that flagellin complexes are unlikely to form in the cytoplasm prior to secretion. We have modified the sentence, which now reads: 'We note that the preferential methylation of surface-exposed lysine residues suggests that the FliB-dependent methylation of lysines depends on some structural feature of flagellin and might occur only after flagellin has at least partially been folded.'

5) Purification of FliC and FliB for mass spectroscopy was not described in method and should be shown.

RE: We apologize for this omission. We have added details concerning purification of FliC and FliB filaments for mass spectrometry analyses.

Other minor comments

line 198

“the surface hydrophobicity So” -> “the surface hydrophobicity (So)”

RE: Changed and thank for spotting this.

line245

The sentence is weird.

RE: We modified the sentence, which now reads: 'Concerning the role of flagella as adhesion molecule, is important to note that the flagellar filament is made of several thousand copies of a single protein, flagellin, which can mediate various interactions with surfaces.'

line253

“Supplementary text S3” -> “Supplementary text S2 and FigS4”

RE: Changed, thanks for spotting this.

line 309 and Table S1

“C2” “C” should be italic.

RE: Changed.

REVIEWERS' COMMENTS:

Reviewer #1 (Remarks to the Author):

The authors have addressed my concerns.

Reviewer #2 (Remarks to the Author):

Horstmann et al have addressed all of my comments for their manuscript titled "Flagella methylation promotes adhesion and host cell invasion of Salmonella Typhimurium". I commend the authors for their improved manuscript which was a pleasure to read. I just have some minor comments for their consideration.

Line 47...change text to "constitutes a novel mechanisms for flagellum-facilitated bacterial adhesion"

Line 71...delete "out"

Line 75...replace "either" with "one"

Line 113...change "dependent on the" to "in" and reference Sup Fig 1

Line 210-219...Suggest that you also describe the FliC-K5R data where you observed reduced adhesion but this mutant had no defect in motility. This data strengthens your findings.

Line 230...delete "the"

Line 284...Add "an" before adhesion molecule. Add "it" before "is important"

Reviewer #3 (Remarks to the Author):

This revised version of the manuscript has fully addressed my concerns and greatly improved. This is a well-executed study and will attract broad interest.

REVIEWERS' COMMENTS:

Reviewer #2 (Remarks to the Author):

Horstmann et al have addressed all of my comments for their manuscript titled "Flagella methylation promotes adhesion and host cell invasion of Salmonella Typhimurium". I commend the authors for their improved manuscript which was a pleasure to read. I just have some minor comments for their consideration.

Line 47...change text to "constitutes a novel mechanisms for flagellum-facilitated bacterial adhesion"

RE: Revised according to editorial changes to abstract.

Line 71...delete "out"

RE: Changed.

Line 75...replace "either" with "one"

RE: Changed.

Line 113...change "dependent on the" to "in" and reference Sup Fig 1

RE: Changed.

Line 210-219...Suggest that you also describe the FliC-K5R data where you observed reduced adhesion but this mutant had no defect in motility. This data strengthens your findings.

RE: Thank you for this suggestion. We now mention the FliC-K5R data as follows: 'Replacing five surface-exposed lysine residues in the D3 domain of FliC with arginine (FliC-K5R) did not affect motility, but affected bacterial surface adhesion as described below (Supplementary Fig. S8a,c).'

Line 230...delete "the"

RE: Changed.

Line 284...Add "an" before adhesion molecule. Add "it" before "is important"

RE: Changed.